# Patient and Healthcare Professional Reflections on Consenting for Extra Bone Marrow Samples to a Biobank for Research—A Qualitative Study

**DOI:** 10.3390/curroncol32030179

**Published:** 2025-03-19

**Authors:** Stuart G. Nicholls, Erika Camilleri, Taryn Chesser, Gary Davis, Katya Godard, Grace Fox, Madeleine Jane Gordon, Krystina B. Lewis, Jocelyn Lepage, Oksana Motalo, Wendy Nuttall, Craig Peleshok, Caryn Y. Ito, Pierre J. A. Villeneuve, Mitchell Sabloff

**Affiliations:** 1Methodological and Implementation Research, Ottawa Hospital Research Institute, Ottawa, ON K1H 8L6, Canada; snicholls@ohri.ca (S.G.N.);; 2Medical School, University of Ottawa, Ottawa, ON K1H 8M5, Canada; 3The Ottawa Hospital, Ottawa, ON K1H 8L6, Canada; 4Independent Researcher, Ottawa, ON, Canada; 5Ottawa Hospital Research Institute, Ottawa, ON K1H 8L6, Canada; 6School of Nursing, Faculty of Health Sciences, University of Ottawa, Ottawa, ON K1N 6N5, Canada; 7Department of Medicine, University of Ottawa, Ottawa, ON K1H 8L6, Canada; 8The Sprott Centre for Stem Cell Research, Ottawa Hospital Research Institute, Ottawa, ON K1H 8L6, Canada; cito@ohri.ca; 9Ottawa Institute of Systems Biology, Ottawa, ON K1H 8L6, Canada; 10Division of Hematology, Department of Medicine, Ottawa Hospital Research Institute, Ottawa, ON K1H 8L6, Canada

**Keywords:** bone marrow, acute leukemia, biobank, qualitative, patient experience

## Abstract

Little is known about patient perspectives regarding consent for obtaining extra research-specific bone marrow (BM) samples during the diagnostic procedure for acute leukemia (AL). This study aimed to better understand patient experiences with consenting to provide these samples and identify potential areas for practice improvement. Semi-structured interviews were conducted with patients treated for AL, 4–6 years prior to the interviews, and healthcare professionals involved with obtaining patient consent and sample collection. A total of 17 patients (14 agreed to provide a sample and 3 did not have a sample in the biobank) and 5 healthcare professionals were interviewed, achieving data saturation. Patients supported increasing public knowledge about research and noted the importance of friends and family in providing emotional support and retaining information. Despite time pressure and anxiety, the decision to donate a research sample did not require much deliberation. Proximal factors informing decisions included impact on patient health and family and anticipated, procedure-associated pain; distal factors included altruism and trust in healthcare professionals. Key information included expected pain and management, the purpose of research samples, and sample security and privacy. Our findings suggest that BM research sample collection may be facilitated through optimizing the environment where information is provided and the type of information provided, including pain management options and the value of the samples for current and future research.

## 1. Introduction

The acute leukemias (AL) [acute myeloid leukemia (AML) and acute lymphoblastic leukemia (ALL)] are cancers of the blood and bone marrow (BM) with rapid onset and progression that necessitate prompt diagnosis and treatment initiation [1,2]. Therefore, upon suspicion of the disease, patients are often admitted to hospital for rapid assessment and diagnosis. Shortly after admission, a BM sample is collected for diagnostic purposes, which is followed by treatment initiation. Despite advances in AL management, many patients still have poor outcomes, with an overall survival of 30–40% [3]. Prognosis is especially poor in the subpopulation of patients with refractory leukemia who do not respond to chemotherapy or relapse within 6 months of achieving a complete remission. Currently, the mechanisms underlying refractory leukemia are incompletely understood. Consequently, research is focused on studying BM samples from patients with newly diagnosed AL to identify potential biomarkers that could predict the sensitivity of AL to chemotherapy [4,5,6]. These studies depend entirely on acquiring research samples at the time of a patient’s diagnostic BM prior to treatment initiation, in order to capture the disease’s underlying biology prior to the influence of therapy.

BM sample collection involves a two-step process: (1) aspiration of multiple liquid marrow samples and (2) collection of a biopsy of the bone from the posterior iliac spine of the pelvis. At the time of the routine diagnostic BM procedure, additional research samples (i.e., other than those for clinical purposes) can be collected and stored in a biobank for research and/or used for current research. Research samples are samples that are separate from the samples obtained for diagnostic purposes. Diagnostic samples are part of the clinical record, identifiable, under the stewardship of the clinical laboratory, and have a retention period. In contrast, research samples are usually obtained after informed consent, stored in a biobank, are not part of the clinical record, de-identified, and are used under the oversight of the local research ethics committee [7,8,9]. While patients with cancer are generally positive towards contributing to medical research, the time constraints imposed by the urgency of the AL diagnosis and treatment initiation raise potential ethical and practical challenges to research sample collection, given the limited number of patients from whom to seek consent, the limited time for deliberation, and the fact that not all samples collected have useful material [10]. Any research sample that is not collected is a huge loss to studies requiring biological samples that could lead to improved patient outcomes. However, ethics standards recognise that individual patient rights are not to be overridden for the benefit of science. There are a lack of studies examining the experience of patients, family members, caregivers, and healthcare professionals with the informed consent process for donating extra, research-specific BM samples, particularly under the time constraints posed by AL. Understanding these experiences and possible areas for improvement would help inform strategies to facilitate research participation among interested patients, while also protecting patient rights and autonomy.

The present study aimed to better understand the experiences and decisional needs—defined as factors that may impact one’s ability to make informed, values-based decisions—of patients and family members regarding consent to donate BM samples for research [11]. Specific research questions were as follows:(1)What are the experiences of patients, families, and caregivers regarding the current approach to consent for extra research-specific BM samples?(2)What are the experiences of healthcare professionals and research staff when approaching patients, families, and caregivers for consent for extra research-specific BM samples?(3)What factors influence patient, family, and caregiver decisions regarding whether to consent for extra research-specific BM samples?(4)What are the informational needs of patients, families, and caregivers regarding the consent process for extra research-specific BM samples?(5)How might the current approach to consent for extra research-specific BM samples be improved?

## 2. Materials and Methods

We conducted a qualitative descriptive study, which was designed in consultation with three patient partners (CP, WN, GS) with lived experience of AL and diagnostic and/or research BM sample collection. We conducted semi-structured interviews with patients approached to provide consent to obtain an extra, research-specific BM sample and healthcare or research staff who were involved in the consent process. We report the study using the SRQR reporting guidelines [12]. We report details of our patient and public involvement using GRIPP2-SF (Appendix A). The study was approved by the Ottawa Health Sciences Research Ethics Board (Ref: 20220762-01H).

### 2.1. Sample and Sampling Frame

Patient lists were reviewed in spring 2023 to identify potential interviewees. Surviving patients who were diagnosed with AL, after being urgently admitted to hospital to confirm the diagnosis, between January 2017 and December 2021 were eligible to be interviewed. This timeframe allowed for a sufficient sample of eligible patients, while also allowing for treatment to be completed. Patients were excluded if they were aware of their diagnosis prior to admission, did not receive intensive induction therapy, were not able to communicate in English or French, or were unable to give consent. Sampling was designed to achieve data saturation, which occurs when additional interviews ceased to provide new information, with an initial estimated sample size of 24–40 participants [13,14,15,16].

### 2.2. Identification and Recruitment

Eligible patients were identified by cross-referencing surviving patients from The Ottawa Hospital’s leukemia database, which contains patient information from The Ottawa Hospital’s electronic medical records system (EPIC, [Verona, WI, USA]), and the list of patients registered in The Ottawa Hospital Hematology Biobank. Patients appearing in both lists were noted as those willing to provide research samples. Unfortunately, declining to provide research samples is not explicitly recorded in either database. Thus, patients not being in the biobank list does not imply that they were unwilling. There could be multiple reasons why a patient was not enrolled in the biobank, such as the patient not being approached, being unsure or hesitant to consent, or declining. Additionally, due to the COVID-19 pandemic restrictions, some research staff were prohibited from approaching patients in-hospital for approximately 6 months. Furthermore, patients who provided consent may have been excluded from the biobank registry if no sample was collected, for technical reasons (e.g., insufficient sample, procedure aborted due to discomfort). We called patients both with and without samples in the biobank to ask if they were willing to be interviewed regarding their experiences surrounding the diagnostic BM procedure. Patients were selected based on leukemia prognostic parameters that were hypothesised to potentially impact their experience, including age at diagnosis, white blood cell (WBC) count at diagnosis (<100 vs. ≥100), year of diagnosis, and sex (gender was not captured in the records at this time). Individuals from each parameter-based group were randomly selected to be approached for interview. Additionally, healthcare professionals and/or research staff who were involved in the consenting process for BM aspirations were identified and asked if they were willing to be interviewed regarding their experiences and perceptions of patient decisional needs.

### 2.3. Data Collection

Two team members (GF and SN) with training and experience in qualitative research methodology conducted the semi-structured interviews. Interviews followed one of two guides: one for patients and one for healthcare professionals or research staff members. Both guides were developed with our patient partners and clinical team. The patient interview guide was piloted with patient partners for clarity and to estimate the likely interview time. The healthcare professionals/research staff guide was piloted with clinical staff among the research team. Following pilot testing, the guides underwent minor revisions for clarity, without substantial change to topic areas.

Interviews were conducted in person or via videoconferencing software depending on interviewee preference. With consent, interviews were recorded and automatically transcribed using videoconferencing software. If consent for recording was declined, written notes were taken to capture key issues, although verbatim quotes could not be obtained. Transcripts were reviewed and edited for accuracy by one member of the research team (GF, SN, or MG). During review, all transcripts were de-identified, with interviewees being assigned unique ID numbers. Final transcript copies were imported into NVivo 13 qualitative data analysis software (Denver, CO, USA) to assist with coding [17].

### 2.4. Data Analysis

Transcripts were analysed using a thematic analysis approach, where textual data were coded and labelled in an inductive manner [18,19]. Interviews were initially coded independently in batches of 2–3 by two members of the research team (GF and SN). Following coding, both members met to review the coding and discuss discrepancies, throughout which a code book was maintained and updated following each consensus meeting. The open coding process was conducted alongside the interview process, using the constant comparison method. Following the open coding phase, individual codes were grouped into overarching themes through data reduction. As coding developed, along with the formation of initial themes, these were presented to the broader research team, including patient partners, for further feedback and interpretation.

### 2.5. Patient and Public Involvement

Three patient partners (CP, WN, GS) were engaged in the project from inception. Meeting monthly, the research team worked with the patient partners to create the research protocol, craft the interview guide, identify potential factors impacting patient decisions, and select parameters for the sampling approach. Patient partners received regular updates regarding the progress of the study and provided feedback on initial qualitative themes. Specifically, they identified themes and quotes that resonated with them the most as patients and contributed to manuscript revisions.

## 3. Results

A total of 373 patients with a diagnosis of AL were extracted from the AL database between January 2017 and December 2021. Of those patients, 235 died prior to initiating the project. After excluding those who were not eligible from the survivors (n = 138), 58 were registered as having a sample in the biobank and 45 were not.

Twenty-two interviews took place from April to August 2023, with sessions lasting from 22 to 40 min. Seventeen patient (including one family member) and five staff interviews were completed. In two interviews, participants declined audio recording and field notes were taken instead. Of the patient participants (n = 17), most were men (n = 11, 65%), and most had donated their sample to the biobank (n = 14, 82%). Most of the healthcare professionals/research staff interviewed were women (80%). Patient participant demographics are provided in Table 1.

While patients were recruited based on records indicating that they had been approached about the biobank, patient recall of details from discussions of the biobank was limited. Thus, the interview did not require specific recollection of what had occurred, but also explored what patients would have preferred. We report responses to the areas we specifically explored, namely experiences of the consent process for research-specific BM donation, in order to identify areas for improvement, as well as the decisional and informational needs of patients, families, and caregivers. Illustrative quotes are presented in the Appendix A.

Experiences of the consent process for research-specific BM donation

When reflecting on the consent process, three key areas were raised among interviewees: preparation and awareness of research, logistical challenges, and having social supports. Patients discussed preparation in terms of when they should be first told about research, but also in terms of having a general awareness about ongoing research in hospitals and its importance to healthcare. Patients felt that increased public awareness of research would lessen the potential surprise at being asked to provide a BM sample for the biobank:


*“[…] when it’s something so obvious that the research and the [HOSPITAL] are one and it’s not confronting, you don’t even have to make the decision of do I want to be part of it or not? Because it’s obvious… It was just, … for me it was something good because I even had more confidence in the [HOSPITAL] since they were doing research, they are at the very fine level of the knowledge and their field because they are doing research.”*
Patient, interviewee #13

Furthermore, staff indicated that greater awareness and sensitisation to research could improve the consent process and protocol adherence (Quote 1.1). When discussing who would be asking for consent for extra, research-specific BM samples, both patients and staff noted that there may be different healthcare professionals involved, from nurses and physician assistants to lab technicians or research staff. However, essential characteristics included being knowledgeable, able to address questions and concerns, and able to recognise patients as people and not just potential research subjects (Quotes 1.2–1.5).

Both staff and patients indicated that consent was only one part of the collection process and that the actual sample collection was affected by organisational factors. Staff reflections tended to focus on coordination and challenges posed by their specific roles and schedules. Additionally, the distance between patients’ locations in-hospital and the hematology ward was described to cause some logistical barriers to timely sample collection. Simultaneously, the staff’s other responsibilities could interfere with their ability to draw the research sample in a timely manner. One staff member described this as follows:


*“[…] for the BM rotations, it just depends how busy it is. Sometimes we’re not just doing BMs all day, like sometimes we’re helping out in the floor, maybe helping clinic. Like sometimes we’re pulled in different directions. So, if I had missed a consent for a biobank, it probably is because I was too rushed to take a look to see if they were supposed to get biobank or not, you know.”*
Staff, interviewee #3

Patients considered how prepared they felt to make this decision amongst all the other information they were receiving. For example, whether they were in a location within the hospital that helped them prepare for the BM sample collection (Quote 1.6). Patients’ reflections commonly referenced the short timeframe from diagnosis to treatment initiation, and the consequent shock and anxiety experienced by patients and family members. One patient described this as follows:


*“[…] we’re just thinking about us, and we want to get better and hopefully we’re going to live, right? So, I don’t think we’re really interested in, you know, some of the people might not be interested in the research part of it, cause it’s just another thing.”*
Patient, interviewee #11

Patients also reflected on the necessity of the sample collection prior to treatment initiation and the benefit of collecting the research and clinical samples simultaneously. While the current consent process offered limited time to consider and prepare for the research sample to be taken, patients noted the major benefit of collecting both samples simultaneously, reducing the number of invasive procedures (Quote 1.7).

Regarding social supports, some participants indicated that a supportive person’s presence was not necessary. However, others indicated the important supportive roles of family members and friends, sometimes distracting them from the situation (Quote 1.8) and offering a ‘second set of ears’ (Quote 1.9) to receive and process the information.

2.Decisional needs and factors influencing patient, family, and caregiver decision-making

Patients noted how shock, anxiety, and information overload regarding their diagnosis may have influenced them to decline providing a sample, even if they were unopposed to research. One patient said the following:


*“If you’ve had a lot of asks on that day, you might have just run out of OK’s. But if you had nothing else asked of you that day, sure, why not? […] I would say also consider the patients state of being if they’re very weak and can’t think straight then either speak to their caregiver or maybe wait till they’re feeling better.”*
Patient, interviewee #2

Patients explained that a request for research consent need not be a one-time occurrence. The fact that they had declined in a particular moment did not necessarily indicate permanent refusal. They were open to being approached again at another time:


*“I can see if they caught me in a bad day I’d be like “no, not today, just not today. Come back, come back another day. The door is closed. Goodbye.”*
Patient interviewee #12

Despite the time pressure and anxiety, the decision whether to give a research sample did not require much deliberation (Quotes 2.1–2.2). This was supported by comments from staff (Quote 2.3), which suggested that many patients reached an immediate decision (Quote 2.4).

Factors affecting patient decisions could be grouped into proximal or more immediate (and individual) factors and those that were more distal or abstract. A key proximal factor was the impact that giving a research sample would have on the patient’s health or family. For example, one patient indicated that they wanted to know if the sample could indicate whether the disease was familial (Quote 2.5), whereas others were concerned about procedure-associated risks, such as infection (Quote 2.6). However, the issues of greatest importance to patients were anticipated procedure-associated pain or whether a sufficient quantity could be collected (Quotes 2.7–2.9). One patient described this as follows:


*“I would say like once you’re poking around, might as well take a little bit more for research. No problem, but no, I wouldn’t have gone through that process twice if there was any way against it.”*
Patient, interviewee #18

Indeed, no patient indicated an intention to decline providing a research sample. When a sample was not given, it was due to challenges with the procedure, such as intolerable pain or inability to retrieve a sufficient sample.

A more distal or abstract reason that influenced patients’ decision to provide a sample was altruism. Some patients reflected on the fact that they were receiving life-saving treatment because of previous patients who had contributed to research (Quotes 2.10–2.13). Others noted that by contributing to research, something positive could come from their diagnosis:


*“I think it gave me a bit of, I don’t know, like hope or goodness. In what I was going through and like obviously it’s not a great experience, but I kind of felt like I could help someone, even though I had to do this, I didn’t have to do it like just for nothing or not just for nothing to save my life, but like, I could like help something in the future.”*
Patient, interviewee #9

Additionally, patients discussed the trust they had in their healthcare providers and how they considered research participation as a means of reciprocating their clinical care (Quote 2.14). One patient noted the following:


*“Everybody was so good that it would be hard to say no to that team. When you’re sick, it’s different. You just want to do whatever you can for the people that are doing everything they can for you. So that’s the way I see it… So yeah, whenever you have a relationship like that with your healthcare professionals and you’ve asked them for so much when they ask back for one thing, you’re not going to say no. At least I’m not going to say no.”*
Patient, interviewee #2

This trust was garnered through demonstrated competency (Quote 2.15), as well as through treating patients holistically and recognizing their needs (Quote 2.16).

3.Information needs and preferences

When asked what information was or would be helpful, there was strong overlap with the decisional needs identified above. Patients wanted to know the number of samples needed, the anticipated pain level, and about possible pain management (Quotes 3.1–3.5).

Other important information included discussion of the research itself and its purpose (Quotes 3.6–3.8). This did not need to be in great detail, but would help justify why the sample was needed:


*“About the research, I think like yeah, kind of having it, not necessarily all of it geared towards it, but like kind of showing what this sample can help with. And like what it can lead to or what it’s working on and stuff like how it’s contributing to better things I think would be a really neat thing to read as a patient, giving this sample kind of like I said, feeling like you’re a part of or doing something good.”*
Patient, interviewee #9

Other information related to privacy, security, and who would access the samples (Quotes 3.9–3.10). It was emphasised that hospital staff should have sufficient knowledge about the biobank (Quotes 3.11–3.12).

Finally, the format of the information provided was discussed. While preferences varied (Quotes 3.13–3.14), a key requirement was that the information should not be burdensome. Patients felt that any information about research should be in a format that would not exacerbate the information overload associated with the early stages of their diagnosis.


*“[…] not too much information. Like if, a pamphlet could be good because it’s usually, it’s really like it’s only one page, so it’s really condensed information […] [It] could be good cause if you, if they told you so or explain you something. Hey, you going to miss it. And with a paper, maybe you’ll be able to read it when you’re going to be more disposed to.”*
Patient, interviewee #12

Notably, patients suggested that documentation—whether printed or electronic—may not specifically inform their decision but could serve as a reference and reminder for the sample’s purpose (Quote 3.15–3.17). Thus, it appeared to serve more as an aide-mémoire than as a decision aid.

## 4. Discussion

In the present study, we explored the experiences of patients diagnosed with AL who were approached for consent to give an extra BM sample for research purposes at the time of their diagnostic procedure, as well as the experiences of the staff involved. Our results point to important practical challenges that may be addressed through organisational changes. We also uncovered new important information that could facilitate more informed decisions and serve as an aide-mémoire for patients to recall why they were being approached to provide a bone marrow sample for research. An interwoven thread throughout the entire discourse was the importance of timing and its significant impact on the decision-making process, which is amplified by the short timeframes from initial suspicion of the diagnosis to research discussions to the confirmatory diagnostic bone marrow procedure. This sequence of events can lead to information overload and a state of shock. Notably, patients highlighted the number of requests that they received during this short period and that, despite an interest in research, they may not have the capacity to deal with another request regarding research. To this end, they pointed to the acceptability of revisiting the consent discussion and that an initial decline may not reflect a disagreement with taking part in research.

Our initial observation that many patients were not able to recall specific details of the consent discussion is consistent with prior studies [20,21,22,23]. Indeed, patients reflected that research was not their focus at the time, indicating it was a lower priority and thus less likely to be recalled. With respect to patient decision-making, our results indicated that the decision to donate was not hugely deliberated, similar to findings from Williams et al.’s study of patients with colorectal cancer, where they expressed “consent is ‘no big deal’ compared with the diagnosis of cancer” [20]. We found that decisions were influenced to a greater degree by immediate practical issues (e.g., anticipated pain/discomfort, infection risk), but also larger issues of altruism and contributing to research. Our findings are strikingly similar to previous work that has consistently identified the important role that contributing to scientific advancement (that they, as patients, have or will benefit from) has on decisions, as well as the importance of helping others with cancer [24,25,26]. However, our work also highlighted the psychological benefits to patients themselves, that contributing to research provided a sense of meaning or a positive aspect to their diagnosis [25]. Altogether, our results reaffirm previous findings by Fradgley et al. that a lack of a research sample may not be due to a refusal of consent [23].

Our findings, however, suggest some concrete changes that could be employed to improve the patient’s comfort in providing consent for the collection of additional research samples. As outlined in Unger et al., patient participation may be limited by institutional barriers, such as ease of access to the clinic or in this case, being admitted to the appropriate ward in the hospital [27]. Therefore, setting the appropriate environment once patients enter the hospital is important. Indeed, the recognition of research as part of care is consistent with ongoing discussions regarding the creation of Learning Health Systems [28]. Studies among cancer patients, and the public more broadly, indicate that despite a willingness to donate research samples to biobanks, there remains a need for improved awareness of biobanks and their role in research [26,29]. This suggests that raising a broader awareness may be beneficial for research generally and specifically for improving biobank sample donations. Further, given the identified importance of time, identifying ways to maximise time for digesting the information in the face of information overload could be an important facilitator in the consent process. Accounting for this in planning might also minimise “decision fatigue” caused by a high volume of requests and new information. Additionally, decision-related pressure on patients may be eased by scheduling a “follow up” discussion with staff after the biobank is first introduced, where patients can reflect before having further opportunity to ask questions and revise their decision if desired. This approach may also help patients to implement some of the other discussed measures to ease decision-making, such as consulting their social support system.

One note of caution, however, relates to comments in which patients discussed a desire to reciprocate for the medical care that they had received—discussed as part of the proposed ‘gift relationship’ [30,31]. While the recognition that their care benefited from research is noted above, and there was a trust in the biobank based on positive relationships with individual representatives, it is important to ensure that patients feel empowered to decline research participation if they feel it is not in their best interest [32]. One potential solution to such concerns would be to have staff not involved in the patient’s care make the approach to discuss research. The ability for the designated individual to answer questions about the biobank was seen as key. As such, having knowledgeable, non-clinical staff members discuss the biobank may be a feasible approach to mitigate concerns regarding coercion.

These findings should be reviewed in the context of the study limitations. One major limitation was that patients and staff were reflecting on events that occurred up to 6 years earlier. While this timeframe was chosen to provide a sufficient sample size and allow some separation between the patient’s care and the current study, it may have affected their ability to recall details from events and circumstances, potentially introducing recall biases. Further, due to this time lag, only a minority of the patients were eligible for the current study, as many did not survive longer than 1–2 years following their AL diagnosis, again potentially introducing biases should those patients with more severe disease (and thus poorer prognosis) have different experiences and preferences than those interviewed. Other important limitations include this study’s single-centre nature, as other centres may employ different processes to approach patients for sample collection, which may affect decisional needs. Furthermore, our study was limited to participants who were able to converse in English or French, potentially excluding important issues that face patients who are unable to converse freely in the dominant languages at a centre. Further work should explore informational and decisional needs among a broader patient cohort.

## 5. Conclusions

In conclusion, as summarized in Table 2, our results suggest that BM sample collection for research may be optimised through addressing barriers to consent at all levels of the process, from structural barriers like the hospital environment to clinical barriers like having trained personnel who can provide information about the use of research samples and the management of anticipated pain. Information for patients should be brief and focus on identified priorities such as procedural pain, justification for research sample collection, and sample storage and access. While the nature of AL requires rapid diagnosis and treatment initiation, opportunities to discuss research do exist and were viewed positively.

## Figures and Tables

**Table 1 curroncol-32-00179-t001:** Baseline demographics.

Item	Number	%
Sex (male)	11	65
Sample in the biobank?		
Yes	14	82%
No	3	18%
Reason for not being in biobank		
Declined	1	
Not asked	1	
Hesitant (provided a sample at relapse)	1	
Diagnosis		
AML	9	53%
ALL	8	47%
Median age at diagnosis (years) [range]	56 [19–68]	
White count at diagnosis (×10^9^/L) [range]	20.3 [1.3–109.7]	

**Table 2 curroncol-32-00179-t002:** Areas for development to improve bone marrow sample collection for research purposes.

Level of Change	Areas of Change
Hospital level	Increase awareness that research occurs as part of healthcare and that one may be asked to participate in research while attending hospital. Develop appropriate systems to minimise risk of missing opportunities to approach patients who may be eligible to donate to the biobank.
Trained personnel	Identify appropriate team members to approach patients and develop training to minimise the risk of coercion. Develop training for team members who are obtaining consent for the samples to ensure they are knowledgeable about the biobank (including structure and uses). Provide clear documentation of the process and about the biobank and which may be retained for future reference by the patient and family members.
Support	Identify opportunities to introduce biobank donation as early as possible to provide the patient and/or family time to digest the material Identify opportunities to allow family or friends to be present to help the patient process the information during a time of information overload.

## Data Availability

Data are contained within this article or the Appendix A.

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
