# Peer review of "Patient and Healthcare Professional Reflections on Consenting for Extra Bone Marrow Samples to a Biobank for Research—A Qualitative Study"

_curroncol, 2025, doi:10.3390/curroncol32030179_

Round 1
Reviewer 1 Report
Comments and Suggestions for Authors
The manuscript deals with an interesting topic, regarding the argumentation extra bone marrow samples donation to biobanks with the purpose of research implementation.
The authors present a qualitative research regarding the donation of extra bone marrow samples by patients affected by onco-hematological diseases and the difficulty of informing patients at a moment in their disease treatment (sometimes potential) in a very early stage of the disease.
However, I believe that, given the extensive literature suggested below, it would be important to consider in the discussion that the issue of informing citizens in general about donation to biobanks is a topic that should be addressed even before the potential onset of the disease.
The long-standing debate on the issue of trust among citizens regarding donation of biological samples to Biobanks, and patients donation to medical research biobanks, should be supported by information policies aimed at increasing people's confidence and trust in research biobanks.
I therefore suggest briefly considering this point in the discussion as well.
Another aspect that the Authors should clarify is whether these extra bone marrow samples are simply residual biological material left after analysis or if such material is insufficient to conduct the proposed research analyses. I believe this point should be specified in the manuscript.
Domaradzk J. et al Perception of Polish patients with cancer of the ethical and legal issues related to biobank research. The Oncologist, 2024, 29, e887–e898.
Domaradzk J. et al. Trust and Support for Cancer Research Biobanks: Insights from Cancer Patients in Poland. Medical Science Monitor 30 (2024).
Tozzo P, et al. Digital Biobanking and Big Data as a New Research Tool: A Position Paper. Healthcare 22;11(13):1825. (2023).
Lensink MA, et al. Better governance starts with better words: why responsible human tissue research demands a change of language. BMC Med Ethics. 23(1):90 (2022)
Dive L, et al. Public trust and global biobank networks. BMC Med Ethics. 2020 Aug 15;21(1):73.
Caenazzo, L et al. Biobanking research on oncological residual material: a framework between the rights of the individual and the interest of society. BMC Med Ethics 14, 17 (2013).
Gefenas E. Turning residual human biological materials into research collections: Playing with consent. Journal of Medical Ethics. 38, 6:351 – 355 (2012).
Reviewer 2 Report
Comments and Suggestions for Authors
The manuscript entitled “Patient and Healthcare Professional Reflections on Consenting for Extra Bone Marrow Samples to a Biobank – a qualitative study, is a well written, easy to follow and understand manuscript. The suggestions, though many obvious, are good ones, to increase the % of patients who would consent for extra bone marrow to be banked for research. I have two suggestions
- Please list in the abstract the following points:
- that the patients that were interviewed were 4 to 6 year post bone marrow biopsy. ,
- that it was patients who had both signed the consent (14) and did not (3).
- and Recognize the small sample size though still generalizable comments .
- In the conclusion , the suggestions of how to increase the number of patients who consent are very good. Please add a table or bullets to make the suggestions and take home messages easier to find.
